# The Multifaceted Landscape of Healthcare-Associated Infections Caused by Carbapenem-Resistant *Acinetobacter baumannii*

**DOI:** 10.3390/microorganisms13040829

**Published:** 2025-04-05

**Authors:** Alessandro Russo, Francesca Serapide

**Affiliations:** Infectious and Tropical Disease Unit, Department of Medical and Surgical Sciences, ‘Magna Graecia’ University of Catanzaro, Viale Europa, 88100 Grosseto, Italy; f.serapide@unicz.it

**Keywords:** carbapenem-resistant *Acinetobacter baumannii*, resistance, severe infections, cefiderocol, prevention

## Abstract

Carbapenem-resistant *Acinetobacter baumannii* (CRAB) is an emerging and important major cause of nosocomial infections, posing a significant challenge to clinicians worldwide. The intrinsic and acquired resistance mechanisms exhibited by CRAB, associated with its ability to persist in healthcare environments, have transformed it into a critical public health concern. The clinical implications of CRAB infections include severe manifestations, like ventilator-associated pneumonia and bloodstream infections. These infections are often associated with increased morbidity and mortality, particularly in critically ill patients, such as those in intensive care units, immunocompromised, and those undergoing invasive procedures. Considering these characteristics, the therapeutic armamentarium for the treatment of CRAB infections is increasingly limited, as these strains exhibit resistance to a broad range of antibiotics, including carbapenems and the new β-lactam inhibitors, which are considered last-line agents for many bacterial infections. An important role is represented by cefiderocol and data from real-world evidence. The aim of this narrative review is to discuss the main topics of CRAB infection and strategies for prevention, management, and therapy.

## 1. Introduction

*Acinetobacter baumannii* is an opportunistic pathogen that poses a significant threat to public health, especially in hospital settings. Its ability to rapidly acquire antibiotic resistance has made it a challenging “superbug” to treat. This bacterium has evolved to rapidly adapt and overcome the challenges posed by a wide range of antibiotics. Many different mechanisms of resistance were described for *Acinetobacter baumannii*, including carbapenemases production, efflux pumps overexpression, porin loss or modification, target site modifications, and antibiotic inactivation.

The relentless rise in antibiotic resistance has precipitated a global health crisis, with carbapenem-resistant *Acinetobacter baumannii* (CRAB) emerging as a particularly formidable adversary. This pathogen, once primarily confined to environmental niches, has progressively established itself as a major cause of nosocomial infections, posing a significant challenge to clinicians worldwide. The intrinsic and acquired resistance mechanisms exhibited by CRAB, coupled with its remarkable ability to persist in healthcare environments, have transformed it into a critical public health concern for healthcare-associated infections [1,2,3]. Of importance, *A. baumannii* can survive in moist environments, contaminating water systems. Therefore, water system management is crucial to prevent contamination. Proper reprocessing of medical devices is equally important to prevent pathogen transmission through contaminated devices.

The clinical implications of CRAB infections are profound, encompassing a spectrum of severe manifestations, including ventilator-associated pneumonia (VAP), bloodstream infections (BSIs), wound infections, and meningitis. These infections are often associated with increased morbidity and mortality, particularly in vulnerable patient populations, such as those in intensive care units (ICUs), immunocompromised individuals, and those undergoing invasive procedures.

Finally, the therapeutic landscape for CRAB infections is increasingly limited, as these strains exhibit resistance to a broad range of antibiotics, including carbapenems, which are considered last-line agents for many bacterial infections [4,5,6]. However, the inherent resistance mechanisms exhibited by CRAB, primarily through the production of carbapenemases, render traditional β-lactam antibiotics ineffective with a predominant role for new drugs like cefiderocol and sulbactam-durlobactam.

The aim of this narrative review is to discuss the main topics of CRAB infection and strategies for prevention, management, and therapy by a review of the more significant literature (last 10 years) about this difficult-to-treat infection.

## 2. Methods

We conducted a search on PubMed from January 2014 to January 2025. The keywords used were “carbapenem-resistant *Acinetobacter baumannii*”; “therapy”; “severe infections”; “resistance”; and “prevention”. We included all trials, case series, case reports and observational, retrospective, or prospective studies. It was considered as a *state-of-the-art review* attempting to summarize the research concerning CRAB infections along the timeline. The objective was to highlight the current state of understanding, and provide an idea of future directions. Based on these considerations, we developed an algorithm for the management and therapy of this infection.

## 3. The Multifaceted Landscape of CRAB Infections

The emergence of CRAB has significantly altered the landscape of healthcare-associated infections, presenting clinicians with a formidable therapeutic challenge. The organism’s remarkable adaptability and resistance to multiple antibiotic classes contribute to the severity and complexity of invasive CRAB infections (see Table 1). These infections typically manifest in patients with underlying comorbidities, prolonged hospital stays, and exposure to invasive medical procedures. Out of the most important invasive infections, urinary tract infections, osteomyelitis with septic arthritis, endocarditis, and peritonitis have an important role in the natural history of CRAB infections (see Table 1).

The clinical spectrum of invasive CRAB infections is diverse, encompassing a range of life-threatening conditions. VAP is a prominent manifestation, particularly in patients requiring prolonged mechanical ventilation. The pathogenesis of CRAB VAP involves the continuous aspiration of secretions, leading to a rapid progression of respiratory distress, fever, and purulent sputum production. The compromised pulmonary function in these patients, coupled with the difficulty in achieving adequate antibiotic concentrations in the lungs, contributes to the high mortality rates associated with CRAB VAP.

BSIs caused by CRAB are another significant clinical entity. These infections often arise from the insertion and maintenance of central venous catheters, providing a direct portal of entry for the pathogen. Clinical presentations of CRAB BSIs can vary widely, ranging from subtle fever and chills to fulminant septic shock with multiple organ dysfunction. The severity of the infection is often influenced by the patient’s underlying health status and the presence of comorbidities.

Wound infections caused by CRAB can significantly impede patient recovery and prolong hospital stays. These infections are particularly concerning in surgical sites and traumatic injuries, where the disruption of skin integrity provides an opportunity for CRAB colonization and subsequent invasion. The clinical characteristics of CRAB wound infections include delayed wound healing, purulent discharge, and surrounding cellulitis. The presence of CRAB in these infections can necessitate extensive surgical debridement and prolonged antibiotic therapy.

Meningitis caused by CRAB, although less common, represents a devastating complication, particularly in patients undergoing neurosurgical procedures or those with head trauma. The ability of CRAB to penetrate the blood–brain barrier and establish infection in the central nervous system underscores its virulence. The clinical manifestations of CRAB meningitis include fever, headache, neck stiffness, and altered mental status. The high mortality rates associated with this infection are attributed to the difficulty in achieving adequate antibiotic penetration into the cerebrospinal fluid and the inherent challenges in managing severe neurological infections.

Several risk factors contribute to the susceptibility of patients to invasive CRAB infections. Prolonged hospitalization, particularly in ICUs, increases the risk of exposure to CRAB in the healthcare environment. The use of invasive medical devices, such as central venous catheters and mechanical ventilators, provides direct portals of entry for the pathogen. Prior exposure to broad-spectrum antibiotics can disrupt the normal microbial flora, creating an opportunity for CRAB colonization and subsequent infection. Immunocompromised patients, including those with hematologic malignancies, solid organ transplants, or AIDS, are particularly vulnerable to CRAB infections due to their weakened immune systems [5,6]. Interestingly, the use of contaminated invasive medical devices, in the scenario of poor clinical practices by healthcare personnel, underlines the role of poor handwashing adherence, poor cleaning and disinfection methods, and lack of isolation of septic patients as important risk factors for invasive CRAB infections.

The pathogenesis of CRAB infections involves a complex interplay of bacterial virulence factors and host immune responses. CRAB’s ability to adhere to surfaces and form biofilms contributes to its persistence in the healthcare environment and its resistance to antibiotics. The production of endotoxins by CRAB can trigger an excessive inflammatory response, leading to sepsis and organ damage. The organism’s diverse resistance mechanisms, including the production of carbapenemases, pose a significant challenge to effective antibiotic therapy [1,2,3].

The diagnosis and treatment of invasive CRAB infections are often complicated by the difficulty in differentiating colonization from true infection. The presence of CRAB in clinical specimens does not always indicate active infection, particularly in patients with indwelling medical devices. The limited availability of effective antibiotics and the rapid emergence of resistance further complicate treatment decision [1,2,3,4].

## 4. Mechanisms of Resistance

*A. baumannii* employs several mechanisms to resist antibiotics, including the following:β-lactamase production: These enzymes inactivate β-lactam antibiotics, such as penicillins and carbapenems.Outer membrane protein alterations: This reduces the permeability of the cell membrane, preventing antibiotics from reaching their targets.Efflux pumps: These systems actively expel antibiotics from the bacterial cell.Target site modifications: Mutations in genes encoding antibiotic targets can prevent drug binding.

One of the most prevalent and insidious resistance mechanisms in *A. baumannii* is the production of β-lactamases. These enzymes act as molecular shields, capable of hydrolyzing the β-lactam ring—a chemical structure common to many antibiotics, including penicillins, cephalosporins, and carbapenems. By disrupting this ring, β-lactamases render the antibiotic incapable of binding to its bacterial targets and inhibiting pathogen growth [7,8,9].

*A. baumannii* excels in β-lactamase production, possessing a repertoire that spans multiple classes of these enzymes. Among the most concerning are carbapenemases, which can inactivate even carbapenems, antibiotics considered last-line treatments for severe infections. The spread of carbapenemase-producing *A. baumannii* strains has severely complicated treatment, often leaving only a few therapeutic options.

The outer membrane of *A. baumannii* is a complex structure that serves as a protective barrier against the external environment, including antibiotics. Alterations to this membrane can reduce permeability, preventing antibiotics from penetrating the bacterial cell and reaching their targets.

One mechanism of outer membrane alteration involves the loss or modification of porins, proteins that form channels through which antibiotics can enter the cell. Reducing the number or size of porins can restrict antibiotic influx, rendering the bacterium less susceptible to their effects.

*A. baumannii* also possesses efflux pumps, transport systems that actively expel antibiotics from the bacterial cell. These pumps can recognize and remove a broad spectrum of antibiotics, contributing to resistance against multiple drug classes simultaneously [10,11].

The hyper-expression of efflux pumps is a common resistance mechanism in *A. baumannii*. Increasing the number or activity of these pumps can lower intracellular antibiotic concentrations, rendering them ineffective.

Antibiotics function by binding to specific targets within the bacterial cell, such as enzymes or cell wall components. Mutations in the genes encoding these targets can alter the structure of the binding site, preventing antibiotic binding and action.

*A. baumannii* can acquire mutations in genes encoding targets of various antibiotics, including enzymes involved in cell wall or DNA synthesis. These mutations can confer resistance to specific antibiotic classes [7,8,9].

*A. baumannii*’s ability to rapidly acquire antibiotic resistance is facilitated by its genomic plasticity. This bacterium can acquire and transfer resistance genes through various mechanisms, including conjugation, transduction, and transformation.

The presence of mobile genetic elements, such as plasmids and transposons, facilitates the spread of resistance genes among *A. baumannii* strains. This ability to exchange genetic material contributes to the rapid emergence and dissemination of multidrug-resistant strains [7,8,9,10].

Finally, ST2 high-risk clone is responsible for a high number of difficult-to-treat infections, but also other clones, like ST636, can carry different *bla* genes. Since certain virulence determinants were present in all strains of both ST2 and ST636 of *A. baumannii*, a detailed detection of beta-lactamase production may be crucial to understand the potential risk of invasive infections [8,9,10].

In summary, *Acinetobacter baumannii* is a formidable adversary, capable of evading antibiotic action through a diverse and sophisticated array of resistance mechanisms. Understanding these mechanisms is crucial for developing novel therapeutic strategies and combating the spread of this multidrug-resistant pathogen (see Table 2).

Of interest, resistance to cefiderocol [12,13,14,15] can arise through several mechanisms, including the following:Mutations in Iron Transport Genes: Cefiderocol leverages the bacterial cell’s iron transport proteins for entry. Mutations in these genes can alter the protein structures, preventing cefiderocol from binding and being transported into the cell. This resistance mechanism is particularly insidious, as it exploits the iron transport system, which is essential for bacterial survival.Increased Efflux: Although cefiderocol circumvents some efflux pumps, the overexpression of others can still contribute to resistance. These pumps can expel cefiderocol from the cell, reducing its intracellular concentration and rendering it ineffective.β-Lactamase Alterations: While cefiderocol is designed to resist hydrolysis by certain β-lactamases, mutations in these enzymes can broaden their activity spectrum, enabling them to inactivate cefiderocol as well.Porin Loss or Modification: Even though cefiderocol uses an iron transport mechanism, porins still play a role in antibiotic entry. The loss or modification of porins can reduce outer membrane permeability, limiting cefiderocol’s access to the iron transport system.

## 5. Current Treatment Options

CRAB infections pose a significant challenge in clinical practice, necessitating the exploration of alternative therapeutic strategies. The inherent resistance mechanisms exhibited by CRAB, primarily through the production of carbapenemases, render traditional β-lactam antibiotics ineffective [16,17,18,19,20,21,22,23,24,25,26,27,28,29,30,31,32,33,34,35,36,37,38,39,40,41,42].

Colistin, a polymyxin antibiotic, remains a cornerstone in the treatment of CRAB infections. Its mechanism of action involves interaction with the lipopolysaccharide (LPS) component of the bacterial outer membrane, leading to membrane disruption and cell death. However, colistin is associated with significant nephrotoxicity and neurotoxicity, necessitating careful monitoring of renal function and neurological status. Furthermore, the emergence of colistin resistance, mediated through modifications in LPS or overexpression of efflux pumps, further complicates its clinical utility. Interestingly, colistin was tested in combination with rifampin, considering colistin’s effect to the outer membrane of Gram-negative bacteria, which causes increasing penetration of rifampin into the bacterial cell.

Tigecycline, a glycylcycline antibiotic, exhibits activity against some CRAB strains. However, its efficacy is limited by the emergence of resistance and suboptimal tissue penetration, particularly in pulmonary infections. Additionally, tigecycline has been associated with an increased risk of mortality in certain patient populations, warranting judicious use. Tigecycline is often employed in combination with other antibiotics, such as colistin or sulbactam, to enhance antimicrobial activity and mitigate resistance development.

Sulbactam, a β-lactamase inhibitor, demonstrates intrinsic activity against *Acinetobacter baumannii*, even in the presence of carbapenem resistance. Its mechanism of action involves the inhibition of penicillin-binding proteins, essential enzymes involved in bacterial cell wall synthesis. Sulbactam is frequently used in combination with other antibiotics, such as colistin or tigecycline, to treat CRAB infections. Combination therapy aims to enhance antimicrobial activity and reduce the risk of resistance.

Given the increasing prevalence of antibiotic resistance, combination antibiotic therapy has become a pragmatic approach in the management of CRAB infections. The combination of two or more antibiotics with distinct mechanisms of action can enhance antimicrobial activity and delay resistance emergence. Common combinations include colistin with tigecycline, colistin with sulbactam, or tigecycline with sulbactam. However, the optimal combination and duration of therapy remain areas of ongoing research [16,17,18,19].

Cefiderocol, a siderophore cephalosporin, represents a novel therapeutic option for CRAB infections. Its unique mechanism of action, involving iron transport-mediated entry into the bacterial cell, allows it to circumvent some of the most common resistance mechanisms. Cefiderocol has demonstrated activity against a broad range of CRAB strains, including those resistant to colistin and tigecycline. However, the emergence of cefiderocol resistance is a growing concern, necessitating careful monitoring [22,23,24,25].

The search for alternative therapeutic options for CRAB infections is ongoing. Bacteriophages, viruses that infect bacteria, and immunotherapy, which aims to enhance the patient’s immune response, represent promising strategies. However, the use of bacteriophages and immunotherapy in the treatment of CRAB infections is still in the experimental phase and requires further clinical studies.

In addition to antibiotic therapy, supportive care plays a crucial role in the management of CRAB infections. Supportive care includes infection control measures, respiratory and circulatory support, and nutritional support. Supportive care is particularly important in immunocompromised patients and those with severe infections [19].

In summary, the treatment of CRAB infections presents a complex challenge that necessitates a multifaceted approach. Antibiotic therapy, combination therapy, supportive care, and the exploration of novel therapeutic strategies are essential to improve clinical outcomes in patients with these severe infections [1,2,3,4,5,6].

Of importance, several professional societies have developed guidelines to address this issue, including the Infectious Diseases Society of America (IDSA), the European Society of Clinical Microbiology and Infectious Diseases (ESCMID), and Italian guidelines authored by Tiseo et al. [16,17,18]. This comparative analysis aims to highlight the similarities and differences in their recommendations.

A common thread across all guidelines is the emphasis on infection prevention and control measures as a first-line strategy to limit the spread of CRAB. Rigorous hand hygiene, contact precautions, and environmental decontamination are consistently advocated. Furthermore, all guidelines underscore the importance of antibiotic stewardship programs to minimize the selective pressure that drives resistance.

In terms of antimicrobial therapy, there is a general consensus on the limited role of single-agent therapy for severe CRAB infections. Combination therapy is widely recommended to enhance antimicrobial activity and reduce the risk of resistance development. However, the specific agents and combinations recommended vary across guidelines [19,20,43,44,45,46,47,48,49,50,51,52].

Colistin remains a cornerstone of CRAB therapy across all guidelines, particularly for severe infections. However, there are nuances in the recommendations regarding dosing and administration. The IDSA guidelines provide specific dosing recommendations based on renal function, while the ESCMID and Italian guidelines offer more general guidance. All guidelines acknowledge the potential for nephrotoxicity and neurotoxicity associated with colistin and emphasize the importance of monitoring.

Tigecycline is another agent that is considered in all three guidelines, although its role is somewhat controversial. The IDSA guidelines suggest that tigecycline may be considered as part of a combination regimen, particularly for infections where colistin is contraindicated or not tolerated. The ESCMID and Italian guidelines also acknowledge the potential utility of tigecycline, but they highlight its limitations, such as suboptimal tissue penetration and the emergence of resistance.

Sulbactam, either alone or in combination, is increasingly recognized as a valuable therapeutic option for CRAB infections. All three guidelines acknowledge the intrinsic activity of sulbactam against A. baumannii. The Italian guidelines, in particular, emphasize the potential role of high-dose sulbactam as a component of combination therapy.

The newer siderophore cephalosporin, cefiderocol, is addressed in the more recent versions of the guidelines. All guidelines acknowledge the potential of cefiderocol as a promising agent for CRAB infections, particularly for those strains that are resistant to colistin and other agents. However, they also highlight the need for further clinical data to define its optimal role in therapy.

The role of other agents, such as minocycline and aminoglycosides, is considered in some of the guidelines, but their use is generally limited to specific clinical scenarios. The Italian guidelines, for example, provide recommendations for the use of minocycline in combination with other agents for certain types of CRAB infections. Finally, data from real-world evidence are necessary to understand the place in therapy of sulbactam-durlobactam [46,48,49,50,51].

In summary, while there are some differences in the specific recommendations, there is a general agreement across the IDSA, ESCMID, and Italian guidelines on the core principles of CRAB management. These include the importance of infection prevention and control, antibiotic stewardship, and combination therapy. The choice of specific agents and combinations should be guided by local epidemiology, susceptibility testing, and patient-specific factors (see Table 3) [21,22,23,24,25,26,27,28,29,30,31,32,33,34,35,36,37,38,39,40,41,42,53,54,55,56,57,58,59,60,61,62,63,64,65,66,67,68,69,70,71,72,73,74,75,76,77,78,79,80,81].

## 6. Prevention and Infection Control Measures

Prevention and infection control are crucial for limiting the spread of *A. baumannii* in healthcare settings [82,83,84,85]. Important measures include the following:Hand hygiene: Proper hand hygiene is essential to prevent *A. baumannii* transmission.Contact precautions: Patients infected or colonized with *A. baumannii* should be isolated to prevent pathogen spread.Environmental cleaning and disinfection: Thorough cleaning and disinfection of the hospital environment can reduce *A. baumannii* contamination.Antibiotic stewardship: Overuse and inappropriate antibiotic use contribute to resistance emergence. Prudent antibiotic use is essential to preserve drug effectiveness.

The ability of this opportunistic pathogen to persist in the environment and rapidly acquire antibiotic resistance necessitates the adoption of effective prevention and infection control strategies.

Hand hygiene is the cornerstone of infection prevention. The transmission of *A. baumannii* primarily occurs through direct or indirect contact with contaminated surfaces or colonized or infected patients. Therefore, meticulous hand hygiene, using either alcohol-based hand rub or soap and water, is fundamental to interrupting the chain of transmission. Adherence to the World Health Organization’s “5 Moments for Hand Hygiene” is essential to ensure effective practice.

When a patient is known or suspected to be colonized or infected with *A. baumannii*, contact precautions must be implemented. These precautions involve placing the patient in a single room or cohorting them with other patients infected with the same organism. Healthcare workers must wear disposable gloves and gowns when entering the patient’s room and remove them before leaving [82,83].

*A. baumannii* can persist in the environment for extended periods, contaminating surfaces and equipment. Therefore, rigorous environmental hygiene is crucial to reducing the environmental burden of the pathogen. Regular cleaning and disinfection of environmental surfaces, particularly high-touch surfaces, are essential. The use of disinfectants with proven efficacy against *A. baumannii* is crucial to ensure adequate disinfection, also considering that *A. baumannii* can show resistance to disinfectant, causing hospital outbreaks.

Active surveillance and screening of high-risk patients can help identify asymptomatic carriers of *A. baumannii*. Early identification of colonized patients allows for the timely implementation of infection control measures, preventing the spread of the pathogen.

Judicious antibiotic use is fundamental to preventing the emergence and spread of antibiotic-resistant *A. baumannii* strains. Antibiotic stewardship programs aim to optimize antibiotic selection, reduce unnecessary antibiotic exposure, and promote the de-escalation of antibiotic therapy.

*A. baumannii* can survive in moist environments, contaminating water systems. Therefore, water system management is crucial to prevent contamination. Proper reprocessing of medical devices is equally important to prevent pathogen transmission through contaminated devices.

The ongoing education and training of healthcare workers is essential to reinforce infection control practices and promote adherence. Regular training sessions, performance feedback, and case discussions can contribute to improving adherence to infection control practices [82,83,84,85].

In the event of *A. baumannii* outbreaks, rapid and effective outbreak management is crucial. This includes the prompt identification of cases, implementation of enhanced infection control measures, and thorough outbreak investigation to identify the source and implement targeted interventions.

Ongoing research is fundamental to developing novel strategies for *A. baumannii* infection control. This includes the development of new disinfectants with enhanced activity against *A. baumannii* and the exploration of alternative decontamination methods, such as ultraviolet light disinfection [84,85].

Interestingly, air grilles may serve as MDR reservoirs. Cohort nursing in open cubicles may not completely prevent MDR transmission through air dispersal, prompting the consideration of future hospital design [86].

By implementing these comprehensive prevention and infection control strategies, healthcare settings can effectively reduce the risk of *A. baumannii* transmission and improve patient outcomes (see Table 4) [87].

## 7. Proposed Therapeutic Algorithm for Invasive CRAB Infections

Here, we present a schematic proposal for the management and treatment of severe CRAB infections.


**Step 1: Confirm CRAB Infection**


Evaluate CRAB colonization.Perform microbiological identification and antimicrobial susceptibility testing. Predominant role for fast microbiology.Assess resistance mechanisms (e.g., carbapenemases, efflux pumps).Consider local epidemiology and resistance patterns.


**Step 2: Select Site-Specific Therapy [16,17,18,19,20,21,22,23,24,25,26,27,28,29,30,31,32,33,34,35,36,37,38,39,40,41,42]**

**Infection Site**

**First-Line Therapy According to Guidelines**

**Alternative Therapy According to Recent Real-World Evidence**

**Bloodstream Infections**
Colistin or Polymyxin B + High-dose Sulbactam ± TigecyclineCefiderocol ± Fosfomycin ± Eravacycline(Sulbactam-Durlobactam)
**Ventilator-Associated Pneumonia**
Colistin or Polymyxin B + High-dose Sulbactam ± Meropenem (if MIC ≤ 8 mg/L)Cefiderocol ± Fosfomycin ± Inhaled Colistin or Amikacin ± Eravacycline(Sulbactam-Durlobactam)
**Meningitis**
High-dose IV and Intrathecal/Intraventricular Colistin or Polymyxin B ± SulbactamMeropenem ± Rifampin ± TigecyclineCefiderocol
**Urinary Tract Infections**
Cefiderocol or IV Colistin ± FosfomycinAminoglycosides (Amikacin/Plazomicin) ± Sulbactam
**Wound and Soft Tissue Infections**
High-dose Sulbactam ± Colistin or TigecyclineCefiderocol ± Fosfomycin
**Osteomyelitis and Septic Arthritis**
Colistin or Polymyxin B + High-dose SulbactamCefiderocol ± Tigecycline ± Rifampin
**Endocarditis**
Colistin or Polymyxin B + High-dose Sulbactam ± RifampinCefiderocol ± Tigecycline or Eravacycline ± Fosfomycin
**Peritonitis**
Colistin or Polymyxin B + High-dose Sulbactam ± MeropenemCefiderocol ± Tigecycline or Eravacycline



**Step 3: Optimize Treatment Strategy**


Combination Therapy: Preferred to prevent resistance and enhance efficacy.High-Dose Sulbactam: If susceptibility confirmed.Colistin Dosing: IV loading dose followed by maintenance; inhaled therapy for pneumonia.Cefiderocol: Effective against CRAB, especially MBL-producing strains.


**Step 4: Monitor and Adjust Therapy**


Evaluate clinical response within 48–72 h.Therapeutic drug monitoring (TDM) for polymyxins and aminoglycosides.De-escalate therapy based on susceptibility results and clinical improvement.


**Key Considerations**


The rise in multidrug-resistant A. baumannii has drastically limited treatment options, often necessitating the use of older, potentially toxic drugs like colistin.Combination therapy is increasingly employed, but optimal combinations and durations remain areas of active research.The development of new antibiotics and alternative therapies like bacteriophages is vital to combat this growing threat.Strict infection control practices within healthcare facilities are paramount to prevent the spread of this pathogen.The continued monitoring of resistance patterns is extremely important.

## 8. Conclusions

*A. baumannii infections* pose a significant therapeutic challenge due to increasing antibiotic resistance. Research into new therapeutic options and the implementation of effective prevention and infection control measures are essential to address this global threat.

The emergence and dissemination of CRAB is driven by a complex interplay of factors, including the overuse and misuse of antibiotics, the selective pressure exerted by healthcare environments, and the remarkable adaptability of *A. baumannii*. The ability of CRAB to acquire and disseminate resistance genes through horizontal gene transfer, coupled with its capacity to form biofilms and persist on environmental surfaces, contributes to its successful colonization and transmission within healthcare settings.

Moreover, the economic burden associated with CRAB infections is substantial, encompassing increased healthcare costs, prolonged hospital stays, and the need for expensive and often toxic alternative therapies.

Addressing the challenge of CRAB requires a multifaceted approach, encompassing strategies for infection prevention and control, antibiotic stewardship, and the development of novel therapeutic agents.

Finally, the development of new antibiotics with activity against CRAB is a critical priority. Further research into sulbactam-durlobactam will enable the use of a drug specifically developed for severe CRAB infections.

## Figures and Tables

**Table 1 microorganisms-13-00829-t001:** Invasive infections caused by CRAB.

Type of Infection	Description	Risk Factors
**Bloodstream Infections**	CRAB is a leading cause of hospital-acquired bacteremia, often associated with high mortality.	Prolonged intensive care unit stay, central venous catheters, immunosuppression, prior antibiotic use.
**Ventilator-Associated Pneumonia**	CRAB is a major pathogen in ventilated patients, leading to severe pneumonia.	Mechanical ventilation, prolonged intubation, prior antibiotic exposure, prolonged intensive care unit.
**(including tracheobronchitis)**		
**Meningitis**	CRAB meningitis is rare but associated with neurosurgical interventions.	Neurosurgical procedures, external ventricular drains, head trauma.
**Urinary Tract Infections**	CRAB can colonize or infect the urinary tract, particularly in catheterized patients.	Indwelling urinary catheters, prolonged hospitalization, antibiotic overuse.
**Wound and Soft Tissue Infections**	Often associated with war injuries, burns, or surgical site infections.	Trauma, burns, invasive procedures, diabetes, immunosuppression.
**Osteomyelitis and Septic Arthritis**	CRAB can cause bone and joint infections, especially in trauma-related cases.	Open fractures, orthopedic implants, diabetic foot infections.
**Endocarditis**	CRAB can rarely cause infective endocarditis, mostly in critically ill patients.	Presence of prosthetic heart valves, central lines, IV drug use.
**Peritonitis**	CRAB is an emerging cause of peritoneal infections, particularly in peritoneal dialysis patients.	Peritoneal dialysis, intra-abdominal surgery, intestinal perforation.

**Table 2 microorganisms-13-00829-t002:** Main mechanisms of resistance to CRAB [10,11,12,13,14,15].

Mechanism of Resistance	Description	Examples and Details
**Carbapenemases Production**	Enzymes that hydrolyze carbapenems and other β-lactams, leading to resistance. These enzymes belong to different molecular classes.	- **Class D OXA-type β-lactamases**: OXA-23, OXA-24/40, OXA-51 (intrinsic), OXA-58. These are the most common in CRAB. - **Class B Metallo-β-lactamases (MBLs)**: NDM-1, IMP, VIM (require zinc for activity, hydrolyze carbapenems but not monobactams).
**Efflux Pump Overexpression**	Actively expel antibiotics from the bacterial cell, reducing intracellular drug concentration. Efflux pumps contribute to multidrug resistance (MDR).	- **RND (Resistance-Nodulation-Division) family**: AdeABC, AdeIJK, AdeFGH (expel carbapenems, aminoglycosides, fluoroquinolones). - **MFS (Major Facilitator Superfamily)**: TetA/B (mediates tetracycline resistance). - **SMR (Small Multidrug Resistance) family**: AbeS.
**Porin Loss or Modification**	Decreased expression or alteration of outer membrane proteins (OMPs), limiting antibiotic entry.	- **CarO porin loss or mutation**: Reduces carbapenem penetration, commonly associated with imipenem resistance. - **OmpA, OmpW alterations**: Can further contribute to antibiotic resistance and virulence.
**Target Site Modifications**	Mutations or alterations in bacterial targets reduce drug binding, leading to resistance.	- **Penicillin-binding proteins (PBPs) alterations**: Mutations in PBP2 and PBP3 reduce β-lactam efficacy. - **GyrA and ParC mutations**: Confer resistance to fluoroquinolones (ciprofloxacin, levofloxacin).
**Antibiotic Inactivation**	Enzymes modify or degrade antibiotics before they reach their target.	- **β-lactamases**: Hydrolyze β-lactams (penicillins, cephalosporins, carbapenems). - **Aminoglycoside-modifying enzymes (AMEs)**: Acetyltransferases, phosphotransferases, nucleotidyltransferases modify aminoglycosides (gentamicin, amikacin).
**Biofilm Formation**	Bacteria form a protective matrix, reducing antibiotic penetration and enhancing persistence.	- **EPS (Extracellular Polymeric Substances) production**: Protects bacteria from host immune response and antibiotics. - **Regulated by quorum sensing (QS) systems**: abaI/abaR, influencing resistance and virulence.

**Table 3 microorganisms-13-00829-t003:** Treatment strategy for CRAB infections [1,2,16,17,18,19,20,21,22,23,24,25,26,27,28,29,30,31,32,33,34,35,36,37,38,39,40,41,42,53,54].

Treatment Strategy	Description	Examples and Details
**Polymyxins (Colistin, Polymyxin B)**	Last-resort antibiotics that disrupt bacterial membranes. Used in combination therapy to reduce resistance.	- **Colistin (Polymyxin E)**: Often administered as colistin methanesulfonate. - **Polymyxin B**: Similar to colistin but does not require conversion from a prodrug. - Risk of nephrotoxicity and neurotoxicity.
**Tetracyclines (Tigecycline, Eravacycline)**	Inhibit protein synthesis; effective against MDR *A. baumannii* but limited for bloodstream infections.	- **Tigecycline**: FDA-approved for complicated intra-abdominal and skin infections, but lower serum levels limit its use in bacteremia. - **Eravacycline**: Newer tetracycline with improved activity against CRAB.
**Sulbactam-Based Therapies**	Sulbactam is a β-lactamase inhibitor with intrinsic activity against *A. baumannii*.	- **Ampicillin/Sulbactam**: High-dose regimens may be effective against sulbactam-susceptible strains. - **Sulbactam-Durlobactam**: New combination with enhanced activity, recently approved.
**Carbapenem-Based Combinations**	Used despite resistance, often in synergy with other drugs.	- **Meropenem + Colistin**: Synergistic effect against some CRAB isolates. - **Meropenem + Vaborbactam** or **Imipenem + Relebactam**: Limited activity against CRAB due to OXA-type carbapenemases.
**Other β-Lactam/β-Lactamase Inhibitor Combination**	Some combinations show activity against specific CRAB strains.	- **Ceftazidime/Avibactam**: Limited activity against OXA-type carbapenemase-producing *A. baumannii*.
**Siderophore-cefalosporin**	New mechanism of action using ferric iron transporter system.	- **Cefiderocol**: A siderophore cephalosporin with good efficacy against CRAB, including MBL-producing strains.
**Aminoglycosides (Gentamicin, Amikacin, Plazomicin)**	Bind to the 30S ribosomal subunit, inhibiting protein synthesis. Used in combination therapy.	- **Amikacin**: More active than gentamicin against CRAB. - **Plazomicin**: May be active against aminoglycoside-resistant strains. - Often combined with polymyxins or carbapenems.
**Fosfomycin**	Inhibits bacterial cell wall synthesis. Used in combination therapy for synergy.	- Limited systemic efficacy but used in combination regimens. - More commonly used for urinary tract infections.
**Adjunctive Therapies (Phage Therapy, Immunotherapy, Antimicrobial Peptides)**	Emerging strategies to enhance treatment effectiveness.	- **Phage therapy**: Investigational, strain-specific bacteriophages. - **Immunotherapy**: Monoclonal antibodies targeting *A. baumannii*. - **Antimicrobial peptides**: Under development as alternative therapeutics.

**Table 4 microorganisms-13-00829-t004:** Measures for control of CRAB spread.

Strategy	Description	Examples and Details
**Hand Hygiene**	Proper hand hygiene is crucial to prevent transmission in healthcare settings.	- **Alcohol-based hand rubs** (preferred) or **soap and water** for visibly soiled hands. - Compliance with WHO’s “Five Moments for Hand Hygiene”.
**Contact Precautions**	Isolation and protective measures to limit spread in hospitals.	- **Use of gloves and gowns** when interacting with infected patients. - **Cohorting or single-room isolation** for CRAB-positive patients.
**Environmental Cleaning and Disinfection**	CRAB can survive on surfaces for long periods; rigorous cleaning is essential.	- **Use of hospital-grade disinfectants** (e.g., hydrogen peroxide, chlorine-based agents). - **Focus on high-touch surfaces** (bed rails, IV pumps, doorknobs).
**Surveillance and Screening**	Early detection of colonized or infected patients helps prevent outbreaks.	- **Active surveillance cultures** in high-risk units (ICU, transplant wards). - **Molecular or culture-based methods** to detect CRAB colonization.
**Antimicrobial Stewardship**	Rational use of antibiotics to reduce resistance development.	- **Avoid unnecessary use of carbapenems and broad-spectrum antibiotics.** - **Guidelines-based prescribing** and **de-escalation strategies**.
**Patient and Staff Education**	Training healthcare workers and informing patients about CRAB risks.	- **Regular infection control training** for hospital staff. - **Patient awareness programs** to improve hygiene compliance.
**Medical Equipment Decontamination**	CRAB can persist on medical devices, requiring strict decontamination protocols.	- **Single-use devices when possible** to prevent cross-contamination. - **High-level disinfection or sterilization** of reusable medical equipment.
**Restriction of Patient Transfers**	Limiting movement of colonized/infected patients between facilities to prevent spread.	- **Strict inter-facility communication** when transferring CRAB-positive patients. - **Screening prior to transfer** in outbreak settings.
**Outbreak Management**	Rapid response to CRAB outbreaks to contain and control infections.	- **Immediate isolation and cohorting of affected patients.** - **Environmental sampling** and **whole-genome sequencing** to track transmission.

## Data Availability

No new data were created or analyzed in this study. Data sharing is not applicable to this article.

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
