# Peer review of "The Multifaceted Landscape of Healthcare-Associated Infections Caused by Carbapenem-Resistant Acinetobacter baumannii"

_microorganisms, 2025, doi:10.3390/microorganisms13040829_

Round 1
Reviewer 1 Report
Comments and Suggestions for Authors
Overall, this is a well-written review of the topic of carbapenem-resistant Acinetobacter baumannii, including the clinical manifestation, resistance mechanisms, and treatment options. The review contains sufficient information on various topics and includes new drugs, including newer beta-lactam beta-lactamase inhibitors. Here are some of my comments that I would like the authors to clarify:
- "Immunocompromised patients, including those with hematologic malignancies, solid organ transplants, or HIV infection, are particularly vulnerable to CRAB infections due to their weakened immune systems." - I understand in haematological malignancies patients have neutropenic fever that can lead to CRAB infections. In solid organ transplants, prolonged hospitalization increases the risk of CRAB infection. However, how does HIV infection increase the risk of CRAB infections? The authors should further elaborate on this point.
- All scientific names should be italicized (Line 198-199)
- In Table 3, the row is better written as "Other β-Lactam/β-Lactamase Inhibitor Combinations" instead of "β-Lactam/β-Lactamase Inhibitor Combinations".
- It would be good if the authors could further explain how rifampicin works in Acinetobacter baumannii in the review.
- In the infection control session, the authors mentioned various measures to help prevent the spread of CRAB in the hospital. There is an increasing number of evidence suggesting that CRAB can be spread through air dispersal at least in case reports. What is the view of the authors on this? The authors may consider to include this in the review.
Author Response
Overall, this is a well-written review of the topic of carbapenem-resistant Acinetobacter baumannii, including the clinical manifestation, resistance mechanisms, and treatment options. The review contains sufficient information on various topics and includes new drugs, including newer beta-lactam beta-lactamase inhibitors. Here are some of my comments that I would like the authors to clarify:
- "Immunocompromised patients, including those with hematologic malignancies, solid organ transplants, or HIV infection, are particularly vulnerable to CRAB infections due to their weakened immune systems." - I understand in haematological malignancies patients have neutropenic fever that can lead to CRAB infections. In solid organ transplants, prolonged hospitalization increases the risk of CRAB infection. However, how does HIV infection increase the risk of CRAB infections? The authors should further elaborate on this point.
R: Dear reviewer, we are really grateful for all your comments that improved quality of the manuscript..
About this first comment, we removed HIV and included AIDS.
- All scientific names should be italicized (Line 198-199)
R: we modified as required.
- In Table 3, the row is better written as "Other β-Lactam/β-Lactamase Inhibitor Combinations" instead of "β-Lactam/β-Lactamase Inhibitor Combinations".
R: we modified as required.
- It would be good if the authors could further explain how rifampicin works in Acinetobacter baumannii in the review.
R: we included rationale for rifampin use in combination, especially with colistin, in paragraph 4.
- In the infection control session, the authors mentioned various measures to help prevent the spread of CRAB in the hospital. There is an increasing number of evidence suggesting that CRAB can be spread through air dispersal at least in case reports. What is the view of the authors on this? The authors may consider to include this in the review.
R: this is another important suggestion. We included it with the reference (Wong SC, et al. Air dispersal of multi-drug-resistant organisms including meticillin-resistant Staphylococcus aureus, carbapenem-resistant Acinetobacter baumannii and carbapenemase-producing Enterobacterales in general wards: surveillance culture of air grilles. J Hosp Infect. 2024 Jul;149:26-35. doi: 10.1016/j.jhin.2024.04.011.)
Reviewer 2 Report
Comments and Suggestions for Authors
This literature review manuscript gives an overview about carbapenem resistant Acinetobacter baumannii. Topic of this manuscript is an interesting and important issue, however, some parts in text should be revised.
1) In case of carbapenem resistant A. baumannii the high-risk clones should be described. The major carbapenem resistant A. baumannii high-risk clone is ST2. This should be descibed in the text. Other additional high-risk clones of A. baumannii could be also mentioned.
2) Disinfectant resistance of A. baumannii should be also described in the text. This is among the main features of A. baumannii strains in hospital outbreaks.
3) In table 3. the cefiderocol should be written in a separate row, not in "beta-lactam-beta-lactamase inhibitor combination". Taking into account the specific chemical structure of cefiderocol, it should be in a separate group as "siderophore-cefalosporin".
Author Response
This literature review manuscript gives an overview about carbapenem resistant Acinetobacter baumannii. Topic of this manuscript is an interesting and important issue, however, some parts in text should be revised.
R: Dear reviewer, your comments were really usefull to improve our manuscript. Thank you very much.
- In case of carbapenem resistant A. baumannii the high-risk clones should be described. The major carbapenem resistant A. baumannii high-risk clone is ST2. This should be descibed in the text. Other additional high-risk clones of A. baumannii could be also mentioned.
R: we modified as required
- Disinfectant resistance of A. baumannii should be also described in the text. This is among the main features of A. baumannii strains in hospital outbreaks.
R: we modified as required.
3) In table 3. the cefiderocol should be written in a separate row, not in "beta-lactam-beta-lactamase inhibitor combination". Taking into account the specific chemical structure of cefiderocol, it should be in a separate group as "siderophore-cefalosporin".
R: we modified as required
Reviewer 3 Report
Comments and Suggestions for Authors
Dear Authors,
The manuscript, "The multifaceted landscape of healthcare-associated infections caused by carbapenem-resistant Acinetobacter baumannii," needs very serious revisions before it can be accepted for further publication steps.
The introduction needs to be rewritten and information added. In its current form, it completely fails to introduce the topics covered in the Manuscript.
line 31/32 - please describe briefly by what pathways/mechanisms resistance occurs in A. baumannii
line 32/33 - please give specific examples of what this evolution consists of
line 34/35 - how did CRAB evolve?
line 36/37 - please provide specific epidemiological data on the global crisis as reported by the authors
line 48/49 - please provide specific therapeutic options used to treat A. baumannii infections. What role do carbapenems play? Please relate this specifically to A. baumannii, not "many bacterial infections."
Also, in the Introduction, please include data on infections (including fatal cases) of A. baumannii vs. CRAB
Please expand the last paragraph "the purpose of the review is...." by including at least the information on the years the Authors relied on to write this Article.
line 54 - please add more details about this occurrence
From line 59, the Authors describe infections related to CRAB, but it should be made clear in the Manuscript that these are the most common infections?
Table 1 - there is no need to describe these infections, it would be more relevant to add “who they affect” and then with VAP: long ventilated patients .... Analogous to the other infections. Such a table would be clearer.
lines 119 - 125 - please describe these mechanisms. Please rearrange the mechanisms of resistance in the form of a diagram, graphic. Then the message to the readers will be easier.
lines 132-136, 137-140, 141-144 - please provide citations
lines 176-181 please write in another form e.g. graphic form.
Table 2 - citations instead of in the title of the Table please provide to each resistance mechanism described.
Chapter 4 please divide into subsections for the specific therapeutic options discussed. Please organize these messages to create these subsections.
Throughout the Manuscript, citations are incorrectly inserted. They are missing in many places. Please add literature items in nefarious places, e.g. line 208. But in other places there is also a need.
The authors have omitted the hugely important role of beta-lactamase inhibitor cephalosporins in this chapter. They show activity against resistant strains causing VAP, including CRAB ( https://doi.org/10.3390/antibiotics13050445)
The authors in their Manuscript have too few citations of scientific studies and reviews available in the literature.
Table 3 - literature items please provide in the last column applying them to each treatment option.
Please add a section on "Materials and methods" including search terms and information on databases that were searched by the Authors. Also, please include the years to which the Authors narrowed their search.
lines 300-3009 please write in a different form, such as in the text using bullets: 1, 2, 3...or I, II, III....
Consider combining these points with the descriptions below. Maybe it is worth introducing subsections again?
Step 2 - use the literature as for Tables 2 and 3.
Line 402 - use the abbreviated name of the bacteria "A. baumannii".
Kind regards
Author Response
Reviewer 3
Dear Authors,
The manuscript, "The multifaceted landscape of healthcare-associated infections caused by carbapenem-resistant Acinetobacter baumannii," needs very serious revisions before it can be accepted for further publication steps.
R: Dear Reviewer, thank you very much for all your efforts to improve quality of our manuscript.
The introduction needs to be rewritten and information added. In its current form, it completely fails to introduce the topics covered in the Manuscript.
line 31/32 - please describe briefly by what pathways/mechanisms resistance occurs in A. baumannii
line 32/33 - please give specific examples of what this evolution consists of
line 34/35 - how did CRAB evolve?
line 36/37 - please provide specific epidemiological data on the global crisis as reported by the authors
line 48/49 - please provide specific therapeutic options used to treat A. baumannii infections. What role do carbapenems play? Please relate this specifically to A. baumannii, not "many bacterial infections."
Also, in the Introduction, please include data on infections (including fatal cases) of A. baumannii vs. CRAB
Please expand the last paragraph "the purpose of the review is...." by including at least the information on the years the Authors relied on to write this Article.
line 54 - please add more details about this occurrence
From line 59, the Authors describe infections related to CRAB, but it should be made clear in the Manuscript that these are the most common infections?
R: we modified as required.
Table 1 - there is no need to describe these infections, it would be more relevant to add “who they affect” and then with VAP: long ventilated patients .... Analogous to the other infections. Such a table would be clearer.
R: we decided to expand in some aspects the Table 1 according with your suggestions. However, these informations are reported in the Table yet.
lines 119 - 125 - please describe these mechanisms. Please rearrange the mechanisms of resistance in the form of a diagram, graphic. Then the message to the readers will be easier.
lines 132-136, 137-140, 141-144 - please provide citations
lines 176-181 please write in another form e.g. graphic form.
R: we modified as required. However, we think that a graphic form could be more confusing that a clear Table. We hope that you can understand it. Moreover, we don’t have for this manuscript a specialistic support to create a Figure that may adequate to explain these complicated and specific mechanisms.
Table 2 - citations instead of in the title of the Table please provide to each resistance mechanism described.
Chapter 4 please divide into subsections for the specific therapeutic options discussed. Please organize these messages to create these subsections.
Throughout the Manuscript, citations are incorrectly inserted. They are missing in many places. Please add literature items in nefarious places, e.g. line 208. But in other places there is also a need.
The authors have omitted the hugely important role of beta-lactamase inhibitor cephalosporins in this chapter. They show activity against resistant strains causing VAP, including CRAB ( https://doi.org/10.3390/antibiotics13050445)
The authors in their Manuscript have too few citations of scientific studies and reviews available in the literature.
R: we modified as required including references in the different sections. About the number of references We think that all relevant manuscript, published on High-quality Journals, were discussed. However, we expanded the list of references. About subheadings in Chapter 4, please can you consider that this paragraph was conceptualizaed to avoid a scholastic (and very frequent in published literature) list of antibiotics used for CRAB? We did not discuss a list of antibiotics but We tried to create a preparatory chapter for the proposed therapeutic algorithm.
Table 3 - literature items please provide in the last column applying them to each treatment option.
Please add a section on "Materials and methods" including search terms and information on databases that were searched by the Authors. Also, please include the years to which the Authors narrowed their search.
R: we included the methods section also if it is not mandatory for narrative review. We don’t understant what do you think about Table 3. However, the proposed algorithm try to report a pratical approach to CRAB treatment.
lines 300-3009 please write in a different form, such as in the text using bullets: 1, 2, 3...or I, II, III....
Consider combining these points with the descriptions below. Maybe it is worth introducing subsections again?
Step 2 - use the literature as for Tables 2 and 3.
Line 402 - use the abbreviated name of the bacteria "A. baumannii".
R: we modified as required. About line 300-309: in the text we reported bullets to highlight major points. Also Table 4 is conceptualized for this scope. Also for this chapter, as reported for chapter 4, we don’t think that subsections are useful.
Kind regards
R: Dear reviewer, again thanks for your comments. We modified, where appropriate in our opinion, the manuscript according with your important suggestions.
We hope that you can understand our point of view and the decision to create an original narrative review for many aspects on which you and me discussed.
Best regards
Alessandro Russo
Reviewer 4 Report
Comments and Suggestions for Authors
The purpose of the submitted manuscript is to provide an overview of infections caused by carbapenem-resistant Acinetobacter baumannii. This work is undoubtedly relevant given the problems associated with this microorganism in the hospital setting. I would like to provide some comments regarding the purpose of improving the quality of the manuscript.
- Title: I have concerns about the term "invasive infections" because "healthcare-associated infections" caused by this microorganism have been recognized worldwide. Therefore, I recommend that the authors consider changing the title from "invasive infections" to "healthcare-associated infections."
2.- Introduction: line 40, page 2: In this same context, the authors indicate that infections caused by this carbapenem-resistant bacteria fall into four main categories. It is important that the authors make it clear that these four infections are "healthcare-associated." Many readers of this type of literature will be in the medical field, so it is important to recognize the aforementioned infections in this way.
- Page 2. Lines 60 to 62: I suggest the authors rephrase that statement. It implies that the "pathogenesis" of the microorganism is caused by the aspiration of contaminated secretions, which is a misnomer.
- Lines 88 to 90: The authors acknowledge the use of invasive medical devices, venous catheters, central catheters, and mechanical ventilation as entry routes for these types of microorganisms. It is important that the authors also mention the use of contaminated invasive medical devices after reprocessing and emphasize poor clinical practices by healthcare personnel in the management of patients with healthcare-associated infections, such as poor handwashing adherence, poor cleaning and disinfection methods, and lack of isolation of septic patients, among others. There is diverse scientific literature that highlights the contamination of surfaces and medical devices in critical areas where patients with infections caused by this bacteria are found.
- Line 98 to line 100: The authors indicate that this microorganism is capable of producing endotoxin, but I notice that there is no bibliographic support for this claim. I request that the authors add a reference to support this claim.
- Table 1: According to the authors' initial report, there are four main infections caused by this microorganism. However, Table 1 describes eight types of infections. It is important that the authors clarify that this microorganism causes "four main healthcare-associated infections," but that there may be some exceptions, which are found in Table 1.
- I notice that throughout the document, whenever β-lactamases are mentioned, they use the word beta. I ask the authors to change it to the Latin symbol, which is the universal way this type of enzyme is reported.
- Lines 140 to 155: The authors present interesting information about the resistance mechanisms of this microorganism; however, the aforementioned lines are not supported by bibliographic references throughout this text. I request that the authors provide bibliographic support.
- Table 2 is interesting because it shows the resistance mechanisms of this microorganism , but it lacks any bibliographic support. I ask the authors to provide the corresponding references.
- Lines 184-224. There are no references to this, and only lines 225-228 mention references "1-6." Are these the same references for addressing the treatment of A. baumannii infections? There are antibiotic management guidelines for this microorganism; I suggest consulting them. Or provide bibliographic support for lines 184-224.
- Table 3. This table describes the treatment strategy for infections caused by this carbapenem-resistant microorganism. While the table is interesting because it summarizes the available treatments, there are no references to the source of the information. It is important to add a column listing the references consulted.
- It 's noteworthy that page nine is completely devoid of bibliographic references. As you may know, the review article format requires references throughout the entire document. This is followed by author comments, but based on the references consulted. I recommend that this page be referenced according to the references consulted.
- The authors strongly recommend including the references to table number five, which I also note has no table title.
- In the conclusion section , I see some typos when writing the name of the microorganism of interest on line 368.
- Without a doubt, the conclusion has to be the general appreciation of the manuscript, I consider it adequate but they could reduce it to 50%, a conclusion is defined as a conclusive and short section.
Author Response
The purpose of the submitted manuscript is to provide an overview of infections caused by carbapenem-resistant Acinetobacter baumannii. This work is undoubtedly relevant given the problems associated with this microorganism in the hospital setting. I would like to provide some comments regarding the purpose of improving the quality of the manuscript.
R: Dear reviewer, thank you very much for all your suggestions. We think that now quality of manuscript is improved.
- Title: I have concerns about the term "invasive infections" because "healthcare-associated infections" caused by this microorganism have been recognized worldwide. Therefore, I recommend that the authors consider changing the title from "invasive infections" to "healthcare-associated infections."
R: we modified as required
2.- Introduction: line 40, page 2: In this same context, the authors indicate that infections caused by this carbapenem-resistant bacteria fall into four main categories. It is important that the authors make it clear that these four infections are "healthcare-associated." Many readers of this type of literature will be in the medical field, so it is important to recognize the aforementioned infections in this way.
- we modified as required
- Page 2. Lines 60 to 62: I suggest the authors rephrase that statement. It implies that the "pathogenesis" of the microorganism is caused by the aspiration of contaminated secretions, which is a misnomer.
R: we modified as required.
- Lines 88 to 90: The authors acknowledge the use of invasive medical devices, venous catheters, central catheters, and mechanical ventilation as entry routes for these types of microorganisms. It is important that the authors also mention the use of contaminated invasive medical devices after reprocessing and emphasize poor clinical practices by healthcare personnel in the management of patients with healthcare-associated infections, such as poor handwashing adherence, poor cleaning and disinfection methods, and lack of isolation of septic patients, among others. There is diverse scientific literature that highlights the contamination of surfaces and medical devices in critical areas where patients with infections caused by this bacteria are found.
R: thank you again for another important observation. We modifed the text accordingly.
- Line 98 to line 100: The authors indicate that this microorganism is capable of producing endotoxin, but I notice that there is no bibliographic support for this claim. I request that the authors add a reference to support this claim.
R: we modified as required.
- Table 1: According to the authors' initial report, there are four main infections caused by this microorganism. However, Table 1 describes eight types of infections. It is important that the authors clarify that this microorganism causes "four main healthcare-associated infections," but that there may be some exceptions, which are found in Table 1.
R: thanks also for this other important observations. We clarified in the text
- I notice that throughout the document, whenever β-lactamases are mentioned, they use the word beta. I ask the authors to change it to the Latin symbol, which is the universal way this type of enzyme is reported.
- R: we totally agree and modified as required
- Lines 140 to 155: The authors present interesting information about the resistance mechanisms of this microorganism; however, the aforementioned lines are not supported by bibliographic references throughout this text. I request that the authors provide bibliographic support.
R: we modified as required
- Table 2 is interesting because it shows the resistance mechanisms of this microorganism , but it lacks any bibliographic support. I ask the authors to provide the corresponding references.
R: we modified as required.
- Lines 184-224. There are no references to this, and only lines 225-228 mention references "1-6." Are these the same references for addressing the treatment of A. baumannii infections? There are antibiotic management guidelines for this microorganism; I suggest consulting them. Or provide bibliographic support for lines 184-224.
R: we modified as required
- Table 3. This table describes the treatment strategy for infections caused by this carbapenem-resistant microorganism. While the table is interesting because it summarizes the available treatments, there are no references to the source of the information. It is important to add a column listing the references consulted.
R: we modified as required.
- It 's noteworthy that page nine is completely devoid of bibliographic references. As you may know, the review article format requires references throughout the entire document. This is followed by author comments, but based on the references consulted. I recommend that this page be referenced according to the references consulted.
R: we modified as required.
- The authors strongly recommend including the references to table number five, which I also note has no table title.
R: Table 5 (named Step 2: Select Site-Specific Therapy) is included in the proposed algorithm. We included references as required.
- In the conclusion section , I see some typos when writing the name of the microorganism of interest on line 368.
R: we modified as required.
- Without a doubt, the conclusion has to be the general appreciation of the manuscript, I consider it adequate but they could reduce it to 50%, a conclusion is defined as a conclusive and short section.
R: we modified as required.
Round 2
Reviewer 4 Report
Comments and Suggestions for Authors
The authors satisfactorily addressed the observations and recommendations made regarding the manuscript. The quality of the document is satisfactory. Therefore, I consider it accepted for publication.
Author Response
The authors satisfactorily addressed the observations and recommendations made regarding the manuscript. The quality of the document is satisfactory. Therefore, I consider it accepted for publication.
R: Dear Reviewer, thank you very much!